# Short-Term Impacts of Ambient Air Pollution on Health-Related Quality of Life: A Korea Health Panel Survey Study

**DOI:** 10.3390/ijerph17239128

**Published:** 2020-12-07

**Authors:** Myung-Jae Hwang, Jong-Hun Kim, Hae-Kwan Cheong

**Affiliations:** Department of Social and Preventive Medicine, Sungkyunkwan University School of Medicine, Suwon 16419, Gyeonggi-do, Korea; mj6663@skku.edu (M.-J.H.); kimjh32@skku.edu (J.-H.K.)

**Keywords:** air pollution, quality of life, EuroQol-visual analog scale, Korea Health Panel Survey

## Abstract

Previous studies have demonstrated that ambient air pollution leads to a decrease in mental and physical function. Although studies on the relationship between long-term exposure to air pollution and health-related quality of life have been conducted, the impact of short-term exposure has rarely been reported. This study explored the association between short-term exposure to air pollution and EuroQol-visual analog scale (EQ-VAS) scores, an indicator of health-related quality of life, using repeated measures. We selected 5420 respondents from seven metropolitan cities (Seoul, Busan, Daegu, Incheon, Gwangju, Daejeon, Ulsan) and one province (Jeju) in South Korea who had participated three or more times in the Korea Health Panel survey conducted from 2009 to 2013. A total of 24,536 observations were used. We applied the daily lag effects of air pollutants on the EQ-VAS stratified by sex and age group using the generalized linear mixed model. After controlling confounders, the EQ-VAS scores decreased statistically significantly in males aged 40–49 years, and females aged 50–64 years with chronic disease. The EQ-VAS scores reduced the most to −1.571 (95% confidence interval: −2.307–−0.834) and −1.722 (95% confidence interval: −2.499–−0.944) per interquartile range increment of carbon monoxide in males aged 40–49 years and per interquartile range increment of sulfur dioxide in females aged 50–64 years, respectively. This study provides evidence that short-term exposure to air pollution is related to the discomfort experienced by individuals in their daily lives.

## 1. Introduction

Globally, ambient air pollution, which causes 4.2 million premature deaths per year, is a major risk factor for public health, and interest in the health impacts of air pollution has been increasing because it causes persistent health damage [1]. The health effects of air pollution, including mortality, morbidity, and physiological disorders, have been reported in South Korea; however, air pollution can cause a deterioration in physical and mental function before progressing to morbidity and mortality. These adverse health effects may manifest as mental discomfort in daily life and closely affect health-related quality of life (HRQOL) [2].

In 2000, the Swiss Study on Air Pollution and Lung Diseases in Adults reported a strong correlation (r > 0.85) between the level of annoyance and the level of particulate matter measuring <10 µm in diameter (PM_10_) and nitrogen dioxide (NO_2_) exposure [3]. Furthermore, the EXPOLIS study reported a high correlation between the indoor and outdoor levels of particulate matter measuring <2.5 µm in diameter (PM_2.5_) and NO_2_ and annoyance scores [4]. In 2005, the score of “vitality”, an item of SF-36, significantly decreased with the increasing level of NOx exposure in Japan [5]. In addition, there was a negative correlation between exposure to air pollution and self-recorded life satisfaction [6,7,8,9]. In 2018, a study in South Korea showed that long-term exposure to air pollution worsens the indicators related to mental health [10,11]. However, many studies related to the quality of life evaluated categorized indicators, and only a few studies used the EuroQol-visual analog scale (EQ-VAS), which is the preferred scale for assessing HRQOL [12,13].

The visual analog scale (VAS), which is reliable for measuring the level of health because of its simplicity, ease of care, and suitability for repeated measurements [14], was used as a measure of HRQOL in the study of cancer patients by Priestman and Baum in the 1970s [15]. The HRQOL considers an individual’s perception of wellbeing, which should be regarded as an index of adverse health outcomes [2] and an important indicator of morbidity and mortality during health interventions.

EQ-VAS, an indicator of health level, may decrease according to short-term exposure to ambient air pollution. Here, short-term exposure to air pollution is also called acute exposure. Acute exposure to air pollution is a short contact with hazardous substances in the atmosphere, which lasts from hours to days. Cho et al. (2014) reported that it was a risk factor for the individual’s emotional states such as depression and, in particular, might worsen the health level of subjects with chronic disease [16]. This also suggests that it may decline an individual’s emotional state and reduce the level of health they experienced on the day.

Based on previous studies, in this study, we adjusted for confounders and observed the impact of short-term exposure to ambient air pollution on changes in EQ-VAS scores using data from the Korea Health Panel (KHP) survey, a nationwide survey conducted every year with repeated measurements, from 2009 to 2013. In particular, the changes in EQ-VAS were estimated in susceptible populations with chronic disease.

## 2. Materials and Methods

### 2.1. Data and Subjects

This study used data from the KHP survey, conducted every year with the cooperation of the Korea Institute for Health and Social Affairs and the National Health Insurance Service, from 2009 to 2013 [17]. Based on 7866 households in 2008, the survey collected databases that could be analyzed in-depth for factors affecting health care, such as individual health status, medical visits, and medical expenditure, for the same household every year. In 2009, additional parameters, including smoking, drinking, physical activity, and quality of life, were surveyed. In our study, we recruited subjects aged ≥18 years who had responded to the EQ-VAS and the HRQOL questionnaire surveys from 2009 to 2013. Based on residential addresses, only subjects from seven metropolitan cities (Seoul, Busan, Daegu, Incheon, Gwangju, Daejeon, Ulsan) and one province (Jeju) were included. There were 1178 respondents who participated in the survey once, 820 who participated twice, 827 who participated thrice, 1090 who participated four times, and 3503 who participated five times; the total number of observations was 27,174. Among these, we analyzed the data of 5420 subjects who responded more than three times, and a total of 24,356 observations were used.

### 2.2. Ethical Statements

The study protocols were reviewed and approved by the Institutional Review Board (IRB) at Sungkyunkwan University (IRB No. 2019-03-019).

### 2.3. Air Pollution and Meteorological Data

South Korea consists of seven metropolitan cities and nine provinces during the study period. These 16 major regions consist of 253 health administrative districts; however, only 96 of them had air pollution monitoring stations installed in 2009. We used the air pollution measurement data collected from 2009 to 2013 by the Ministry of Environment, South Korea. The data provided hourly measurements of PM_10_, NO_2_, carbon monoxide (CO), and sulfur dioxide (SO_2_); however, PM_2.5_ were not measured during the study period. The measured data were obtained every hour and refined to calculate the daily mean concentrations.

We used the daily mean temperature and humidity data from the Korea Meteorological Administration. They also provided an hourly measurement of temperature and humidity, measured in 45 sites in the 16 major regions. We classified the measured data according to the 16 regions and calculated the daily mean temperature and humidity. We matched the daily mean air pollution data and meteorological data with the KHP survey data based on the subjects’ survey date and residential address. We applied the lag0 (same day as the date of the survey conducted) to lag3 (3 days before the survey conducted) for the lagged effect of air pollution exposure, and a moving average of 0 to 3 days (lag0–3 days) before the day of the survey conducted. We calculated the change in EQ-VAS scores by applying the interquartile range (IQR) of air pollutants. The model was adjusted by the daily mean temperature and humidity.

### 2.4. Measurement of the Health-Related Quality of Life

We used the Korean version of the EQ-VAS approved by the EuroQol group as an HRQOL tool. The VAS has a high degree of validity and reliability for measuring the quality of life [18,19]. The EQ-VAS was presented as a vertical line, marked from 100 (best imaginable health state) to 0 (worst imaginable health state) in the center of each page, with four profiles presented in boxes to the left and right of it [20,21]. We used the EQ-VAS scores as the dependent variable for the subject’s response to the question of quality of life.

### 2.5. Variables Measurement

We considered sex, age, education level, income, marital status, economic activity, smoking status, alcohol consumption, sleeping time, and the presence of chronic diseases as covariates. We divided the respondents according to their socioeconomic age: <30, 30–39, 40–49, 50–64, and ≥65 years. The level of education was classified as “elementary school”, “middle school”, “high school”, and “university and more”. The income level was categorized as the national income of South Korea in five divisions (first quintile: lower 20%, second quintile: lower 21–40%, third quintile: upper 41–60%, fourth quintile: upper 21–40%, and fifth quintile: the highest 20%). Marital status was classified as “single”, “married”, “divorced”, and “bereaved”. Economic activity was classified as “active” and “non-active”. Smoking status was identified as “no”, “past”, and “current”. Alcohol consumption was classified as “no or less than once a month”, “less than two or three times a week”, and “almost every day”. The subjects’ sleeping hours were classified as “less than 6 h per day” and “6 h or more per day”. We identified the presence of chronic diseases using the linked subjects’ health insurance data. We stratified all subjects by sex, age group and the presence of chronic disease to observe the impact of air pollution on EQ-VAS scores.

### 2.6. Statistical Analysis

Considering the repeated measurements of EQ-VAS scores based on the panel study, we used a generalized linear mixed model (GLMM) that assumes a Gaussian distribution to estimate the effect of air pollution on EQ-VAS scores. To select the most suitable linear mixed model (model 1, model 2, and model 3) (Appendix A) among several models for our data, we used a conditional Akaike’s information criterion (cAIC) based on conditional probability density function that considers the estimation of parameters and a prediction of random effects [22].
cAIC = −2l + 2K(1)
where I is the log-likelihood function of the model considered, and K is the number of parameters of the model. After comparing the cAIC values, we selected model 3, which had the minimum cAIC value, as the most appropriate model for this study. The model 3 specifications are as follows:(2)$$\begin{array}{l} E\left(Y\right)=\beta_{0}+\beta_{1}\left(\mathrm{pollutants}\right)+\beta_{2}\left(\mathrm{temperature}\right)+\beta_{3}\left(\mathrm{humidity}\right)+\beta_{4}\left(\mathrm{sex}\right)+\beta_{5}\left(\mathrm{age}\right)\\ \;\;\;\;\;\;\;\;\;\;\;\;\;\;\;\;\;\;\;\;\;\;\;\;\;\;\;\;\;\;+\beta_{6}\left(\mathrm{education}\right)+\beta_{7}\left(\mathrm{income}\right)+\beta_{8}\left(\mathrm{marital}\;\mathrm{status}\right)+\beta_{9}\left(\mathrm{smoking}\right)\\ \;\;\;\;\;\;\;\;\;\;\;\;\;\;\;\;\;\;\;\;\;\;\;\;\;\;\;\;\;\;+\beta_{10}\left(\mathrm{drinking}\right)+\beta_{11}\left(\mathrm{economic}\;\mathrm{status}\right)+\beta_{12}\left(\mathrm{chronic}\;\mathrm{disease}\right)\\ \;\;\;\;\;\;\;\;\;\;\;\;\;\;\;\;\;\;\;\;\;\;\;\;\;\;\;\;\;\;+\beta_{13}\left(\mathrm{sleeping}\;\mathrm{time}\right)+\gamma \left(\mathrm{subject}\right) \end{array}$$
where E(Y) is the expected score of EQ-VAS, and γ is the random effect for subjects.

We also used a generalized additive mixed model (GAMM) to observe the association between air pollution and EQ-VAS scores. Like GLMM, the linearity relationship was shown by all the adjusted covariates included in model 3. All statistical analyses were conducted using the R version 3.5.3 (The R Foundation for Statistical Computing, Vienna, Austria) with the “lme4” and “gamm4” package for GLMM and GAMM model fitting, respectively. The level of statistical significance was set at *p* = 0.05, and 95% confidence intervals (CIs) were estimated for the point estimates.

## 3. Results

Of the 7418 subjects, 5420 participated in the survey more than three times. The predominant age group was 50–64 years in males (27.7%) and females (26.4%), as shown in Table 1. The education level of most of the subjects was “high school”, and the income level with the highest proportion was in the fifth quintile of respondents. Of all subjects, most were married and economically active. Contrary to males, most females were classified under the “no smoking” category of smoking status and the “no or less than once a month” category of alcohol consumption. Of all subjects, most subjects were sleeping 6 h or more per day in males (85.1%) and females (81.3%). During the study period, 53.3% of male subjects and 62.5% of female subjects had at least one chronic disease.

During the study period, the mean exposure concentrations of PM_10_, NO_2_, CO and SO_2_ were 45.6 μg/m^3^, 24.7, 463.2 and 5.0 parts per billion (ppb), respectively (Table 2). The IQRs of PM_10_, NO_2_, CO and SO_2_ were 26.0 μg/m^3^, 13.8, 162.5, and 3.0 ppb, respectively. In meteorological indices, the mean temperature was 16.6 °C and the relative humidity was 65.1%.

We identified a model to adjust the covariates that affect the EQ-VAS scores. Based on the values cAIC of different models, model 3 with the minimum cAIC was adopted (Appendix A). We stratified all subjects by sex, age group and presence of chronic disease. The EQ-VAS scores per IQR increment in all subjects reduced but was not statistically significant (Table 3). Stratified by sex, the EQ-VAS scores per IQR increment of NO_2_, CO, and SO_2_ decreased significantly in females. When stratified by age group, the most significant decreases were −1.041 (95% CI: −1.683–−0.399) per IQR increment of SO_2_ at 40–49 years old and also reduced by −0.617 (95% CI: −1.190–−0.044) per IQR increment of NO_2_ at 50–64 years old. However, no significant correlation was observed in ≥65 years age group. In subjects with chronic disease, the decrease in EQ-VAS scores due to short-term exposure to air pollution was evident. The EQ-VAS was −0.251 (95% CI: −0.499–−0.004) per IQR increment of PM_10_. When stratified by sex, these effects were more sensitive in females. When stratified by age group, it decreased statistically significantly in the 40–49 years old and 50–64 year old age groups, and generally reduced more significantly in the 50–64 years old age group. It was −1.135 (95% CI: −1.562–−0.709) per IQR increment of PM_10_ and −1.505 (95% CI: −1.999–−1.012) per IQR increment SO_2_.

The spline curve in Figure 1 shows the relationship between air pollution and EQ-VAS scores of subjects with chronic disease using GAMM. The EQ-VAS scores decreased as the exposure to each air pollutant increased. In particular, the EQ-VAS scores rapidly decreased above a certain concentration of air pollution.

We observed the short-term effects of air pollution on EQ-VAS in males aged 40–49 years and females aged 50–64 years with chronic diseases (Figure 2). The shorter the lag effects of air pollution, the greater the decrease in EQ-VAS in all subjects. In Figure 2a, the EQ-VAS per IQR increment of CO decreased the most to −1.571 (95% CI: −2.307–−0.834) at lag0–3 days, and the same pattern was observed in PM_10_, NO_2_, and SO_2_. In Figure 2b, the EQ-VAS per IQR increment of SO_2_ reduced the most, −1.722 (95% CI: −2.499–−0.944), followed by −1.688 (95% CI: −2.366–−1.009) in PM_10_ at lag0–3 days.

## 4. Discussion

We used the KHP survey data from 2009 to 2013 to explore the impact of short-term exposure to ambient air pollution on EQ-VAS scores, which indicate the subjective health status of an individual. After considering socioeconomic and other confounding factors related to HRQOL, we believe that short-term exposure to air pollution is significantly associated with a decrease in EQ-VAS scores. There was a significant difference in the stage of life cycle in adults of each sex, and there was a strong association between the level of air pollutants and EQ-VAS scores in males aged 40–49 years and females aged 50–64 years.

This study provided evidence that was consistent with a previous study. Shin et al. [10] demonstrated that individuals aged <65 years are more vulnerable to deterioration in mental health due to air pollution than those ≥65 years old. Our study also showed a stronger correlation between the level of air pollutants and EQ-VAS scores in males aged 40–49 years who were more economically active than in males aged ≥65 years in South Korea. This could be due to frequent exposure to air pollution because of increased social activities [23,24]. In females, exposure to air pollution during menopause is more likely to have a significant impact on health than frequent exposure to air pollution in other age groups. Consequently, females undergoing menopause are more sensitive to the mental health effects of air pollution than females before menopause [25,26,27]. This is felt as the discomfort caused by exposure to air pollution on the day of the survey and is reflected in the health condition and HRQOL of the subject at the time of the survey. To date, a survey on the effect of menopause on the sensitivity to air pollution has not been conducted, making it difficult to provide a scientific explanation for the significant reduction in the EQ-VAS scores of middle-aged females.

It has been shown that exposure to airborne harmful substances has been shown to trigger inflammatory processes in the respiratory system, leading to systematic inflammatory mediator circulation [28]. These mediators produce antibodies through an immune system response, which can cross the blood–brain barrier, causing neurological inflammation and oxidative stress in the brain. These inflammatory factors play an important role in the development of major depressive disorders through neurochemical changes [29]. This impaired state of mental health can manifest as physical and mental discomfort in daily life.

Although there are many studies on the effects of air pollution on easy-to-measure indicators such as morbidity and mortality, there are no studies on health-related indicators of quality of life that compound them. To the best of our knowledge, few studies have presented results quantitatively using EQ-VAS scores rather than odds ratios. In this study, the EQ-VAS score per 13.8 ppb increment of NO_2_ was significantly reduced by 1.442 in females aged 50–64 years. On the EQ-VAS, measured from 0 to 100, a change of 1.442 can be minimal. However, it can be seen in Figure 1 that EQ-VAS scores are not affected by air pollution below a certain concentration; however, they rapidly decrease above a threshold concentration. Considering that the daily average maximum concentration of NO_2_ during the study period was 73.7 ppb, it was shown that the difference in EQ-VAS experienced by individuals may occur up to 7.701 depending on the variation in air pollution exposure levels between regions. This is subjective, and overall individual discomfort on the day may appear as a bodily and mental condition and represent the health status of the population.

To date, there has been controversy regarding the use of self-reported EQ-VAS as a tool to measure health indicators. The reduction in the EQ-VAS score is an indirect indicator of extreme pain and discomfort [30]. It is not known whether a broader and heterogeneous interpretation of this concept is evident in the population or clinical subgroup. However, in a study where the overall health status is the measurement target, using EQ-VAS is more appropriate than EQ-5D profile data weighted according to general public preference [14].

Our study has a few limitations. First, the observation data of the PM_2.5_ levels before 2013 have not been released by the Ministry of Environment, and thus, the evaluation of exposure to PM_2.5_ could not be conducted. Previous studies have reported that PM_2.5_ is strongly associated with mental health problems [31]. If there were available sources of PM_2.5_ measurement data, a stronger association of HRQOL with short-term exposure to air pollution could have been estimated. Second, the level of air pollution was determined for only eight major regions because of the limited number of air pollution monitoring stations. If a higher resolution of exposure data from more administrative regions had been available, we might have observed more significant changes in the EQ-VAS scores with short-term exposure air pollution. In addition to the outdoor air pollution data used in this study, if indoor air quality data had been obtained, we could have explored the impact of short-term exposure to air pollution on EQ-VAS scores more comprehensively.

The advantage of this study was the use of the KHP survey, which produces annual data on individual health, medical use, and medical expenditure for the same household every year since 2008. A high proportion of subjects (73.1%) participated in the survey for 5 years from 2009 to 2013. Thus, the survey provided time-series data in which the changes in EQ-VAS scores could be observed over time with changes in exposure to air pollution through repeated measurement in the same subject. In particular, the health insurance data of each subject contributed to the systematic management of the quality of the survey and ensured the validity of the research design. Moreover, the study provided a quantitative estimation of the impact of short-term exposure to air pollution using the EQ-VAS. Most studies on air pollution and HRQOL have explored the effects of long-term exposure and presented the binomial outcome of indicators related to the quality of life. We used the survey dates to estimate the impact of short-term exposure to air pollution on EQ-VAS scores and provide evidence of acute rather than chronic health impacts. Furthermore, the changes in the HRQOL associated with acute exposure to air pollution were stratified at the national level according to sex, age, and the presence of chronic disease in subjects.

Globally, the health impact of air pollution remains a public health problem. Currently, numerous studies have observed associations between air quality and various health outcomes, such as mortality, hospitalization for cardiovascular disease, and incidence of respiratory symptoms [32]. However, we believe that air pollution still has negative health impacts other than those reflected in morbidity and mortality measurements [5]. The HRQOL refers to the way an individual feels and physically functions in daily life and deterioration in health. Scientifically validated HRQOL can measure the health status of populations that cannot be estimated by morbidity or mortality [2]. The World Health Organization has broadened the perspective of health to integrate the concept of HRQOL [5,33]. In addition, the American Thoracic Society noted that assessing the adverse health impacts of air pollution based on HRQOL scores is important in defining policies related to personal and public health issues [2]. In other words, through HRQOL data such as EQ-VAS, unseen ambiguous health problems in the population can be made more visible and concrete [5]. Thus, the results of this study can be used as a scientific basis for establishing an air quality improvement policy that considers the health impacts on susceptible populations at the local level.

Our study is a rare report to estimate the relationship between short-term exposure to air pollution and EQ-VAS scores based on a panel study. The concept of air pollution such as “particulate matter” and “fine dust” emerged in South Korea after severe yellow dust was observed in the winter of 2013. Subsequently, a policy to improve the air quality in South Korea was more strictly implemented. Thus, information bias was excluded from the measurement of EQ-VAS in this study since the subjects were surveyed without any knowledge of air pollution concentrations. This suggests that even short-term exposure to air pollution may affect the health of an individual.

## 5. Conclusions

We demonstrated that daily exposure to ambient air pollution is associated with a decrease in EQ-VAS score, an indicator of HRQOL. This impact was more apparent in males aged 40–49 years and females aged 50–64 years with chronic disease. These findings indicate that exposure to harmful substances in the atmosphere might decrease mental and physical function. With a more robust policy to reduce air pollution based on this evidence, improved life satisfaction could be expected in the population to benefit from the policy.

## Figures and Tables

**Figure 1 ijerph-17-09128-f001:**
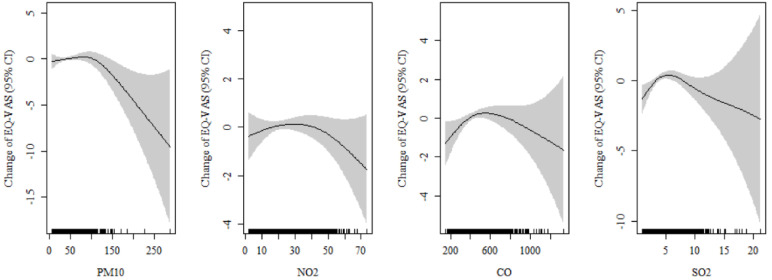
Change in the EQ-VAS scores per unit increment of air pollutants in subject with chronic diseases at lag 0–3 days. The model was adjusted for sex, age, education, income, marital status, smoking, alcohol consumption, economic status, and sleeping time. CI, confidence interval; PM_10_, particulate matter measuring <10 µm in diameter; NO_2_, nitrogen dioxide; CO, carbon monoxide; SO_2_, sulfur dioxide.

**Figure 2 ijerph-17-09128-f002:**
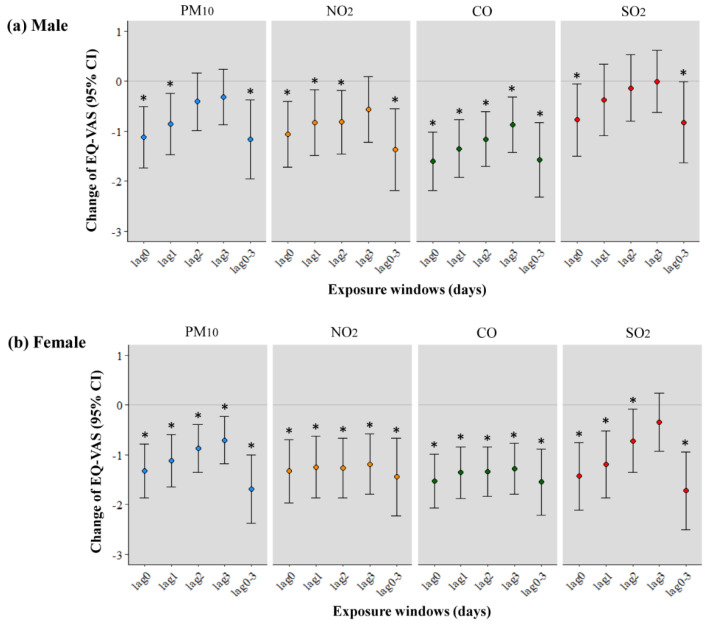
Association between changes in the EQ-VAS scores per interquartile range increment of PM_10_, NO_2,_ and SO_2_ in (**a**) males aged 40–49 years and (**b**) females aged 50–64 years with chronic diseases. * *p*-value < 0.05. The model is adjusted for age, education, income, marital status, smoking, alcohol consumption, economic status, and sleeping time. PM_10_, particulate matter measuring <10 µm in diameter; NO_2_, nitrogen dioxide; SO_2_, sulfur dioxide; IQR, interquartile range; CI, confidence interval; lag0–3, moving average 0 to 3 days.

**Table 1 ijerph-17-09128-t001:** Demographic characteristics of the study participants by sex in the Korea Health Panel data from 2009 to 2013 (total *N* = 27,174).

Variables	Male	Female	
	*N* (%)	*N* (%)	*p*-Value ^†^
	12,197 (44.9)	14,977 (55.1)	
Age group (years)			<0.001
<30	1593 (13.1)	2173 (14.5)	
30–39	2017 (16.5)	2609 (17.4)	
40–49	2832 (23.2)	3170 (21.2)	
50–64	3376 (27.7)	3953 (26.4)	
≥65	2379 (19.5)	3072 (20.5)	
Education level			<0.001
Elementary school	1455 (11.9)	3376 (22.5)	
Middle school	1206 (9.9)	1898 (12.7)	
High school	5008 (41.1)	5319 (35.5)	
University and over	4528 (37.1)	4384 (29.3)	
Income quintile			<0.001
First quintile	1213 (9.9)	1859 (12.4)	
Second quintile	2197 (18.1)	2774 (18.5)	
Third quintile	2694 (22.1)	3235 (21.6)	
Fourth quintile	2933 (24.0)	3343 (22.3)	
Fifth quintile	3160 (25.9)	3766 (25.2)	
Marital status			<0.001
No	2594 (21.3)	2622 (17.5)	
Married	8968 (73.5)	9995 (66.7)	
Divorced	432 (3.5)	488 (3.3)	
Bereaved	203 (1.7)	1872 (12.5)	
Economic activity			<0.001
Yes	8943 (73.3)	7023 (16.9)	
Smoking status			<0.001
No	2881 (23.6)	14,338 (95.7)	
Past	4033 (33.1)	298 (2.0)	
Current	5283 (43.3)	341 (2.3)	
Alcohol consumption			<0.001
No or less than once a month	4291 (35.2)	10,862 (72.5)	
Less than two or three times a week	6543 (53.6)	3970 (26.5)	
Almost every day	1363 (11.2)	145 (1.0)	
Sleeping hours			<0.001
<6 h per day	1816 (14.9)	2794 (18.7)	
≥6 h per day	10,381 (85.1)	12,183 (81.3)	
Presence of chronic disease			<0.001
Yes	6495 (53.3)	9368 (62.5)	
Number of participations			<0.001
Once	628 (17.9)	550 (14.1)	
Twice	458 (13.0)	362 (9.3)	
Three times	436 (12.5)	391 (9.9)	
Four times	614 (17.5)	476 (12.2)	
Five times	1376 (39.1)	2127 (54.5)	☐

† *p*-values were obtained by comparing the groups using the chi-square test of Fisher’s exact test.

**Table 2 ijerph-17-09128-t002:** Exposure level of air pollutants and meteorological indices during the study period.

				Percentile				
Exposure	Mean	SD	Min	25th	50th	75th	Max	IQR
Daily exposures								
PM_10_ (μg/m^3^)	45.6	22.1	4.9	30.3	41.8	56.3	278.9	26.0
NO_2_ (ppb)	24.7	11.8	2.0	16.3	22.7	30.1	73.7	13.8
CO (ppb)	463.2	152.1	150	362.5	436.4	525	1176.0	162.5
SO_2_ (ppb)	5.0	2.3	1.0	3.3	4.5	6.3	21.3	3.0
Temperature (°C)	16.6	8.1	−8.0	9.5	18.7	23.2	31.1	13.7
Relative humidity (%)	65.1	16.4	20.0	55.0	66.0	78.0	99.0	23.0

PM_10_, particulate matter measuring <10 µm in diameter; NO_2_, nitrogen dioxide; CO, carbon monoxide; SO_2_, sulfur dioxide; ppb, parts per billion; SD, standard deviation; IQR, interquartile range.

**Table 3 ijerph-17-09128-t003:** The change in EuroQol-visual analog scale (EQ-VAS) scores per interquartile range increment of air pollutants in all subjects and subjects with chronic disease stratified by sex and age group.

	Groups	EQ-VAS (95% CI) ^a^			
		PM_10_	NO_2_	CO	SO_2_
All subjects	All	−0.110 (−0.314–0.094)	−0.101 (−0.342–0.141)	−0.156 (−0.363–0.050)	−0.087 (−0.349–0.175)
	With chronic disease	−0.251 (−0.499–−0.004)	−0.165 (−0.567–0.236)	−0.314 (−0.660–0.031)	−0.158 (−0.564–0.248)
Sex					
Male	All	−0.176 (−0.478–0.126)	−0.052 (−0.403–0.298)	−0.129 (−0.435–0.178)	−0.031 (−0.419–0.358)
	With chronic disease	−0.228 (−0.530–0.074)	−0.211 (−0.562–0.139)	−0.170 (−0.459–0.119)	−0.063 (−0.449–0.323)
Female	All	−0.157 (−0.415–0.102)	−0.191 (−0.518–0.136)	−0.809 (−1.087–−0.530)	−0.130 (−0.484–0.225)
	With chronic disease	−0.260 (−0.536–0.016)	−0.308 (−0.627–0.010)	−0.927 (−1.195–−0.660)	−0.144 (−0.553–0.265)
Age group (years)					
<30	All	−0.485 (−1.034–0.063)	−0.403 (−1.101–0.296)	−0.513 (−1.112–0.086)	−0.207 (−0.646–0.232)
	With chronic disease	−0.537 (−1.086–0.011)	−0.541 (−0.239–0.158)	−0.676 (−1.275–−0.076)	−0.682 (−1.401–0.037)
30–39	All	−0.298 (−0.737–0.142)	−0.230 (−0.768–0.309)	−0.358 (−0.817–0.101)	−0.356 (−0.882–0.169)
	With chronic disease	−0.558 (−0.998–−0.118)	−0.372 (−0.912–0.167)	−0.967 (−1.428–−0.506)	−0.656 (−1.182–−0.131)
40–49	All	−0.355 (−0.782–0.071)	−0.395 (−0.856–0.066)	−0.978 (−0.769–0.012)	−1.041 (−1.683–0.399)
	With chronic disease	−1.135 (−1.562–−0.709)	−0.671 (−1.132–−0.210)	−0.850 (−1.241–−0.460)	−0.927 (−1.366–−0.488)
50–64	All	−0.211 (−0.711–0.290)	−0.617 (−1.190–0.044)	−0.319 (−0.809–0.171)	−0.245 (−0.739–0.248)
	With chronic disease	−1.015 (−1.414–−0.617)	−1.060 (−1.514–−0.606)	−1.194 (−1.574–−0.814)	−1.505 (−1.999–−1.012)
≥65	All	0.116 (−0.376–0.609)	−0.059 (−0.615–0.496)	−0.319 (−0.778–0.139)	0.007 (−0.626–0.639)
	With chronic disease	−1.116 (−0.507–0.274)	−0.059 (−0.304–0.186)	−0.337 (−0.794–0.119)	−0.447 (−1.009–0.116)

^a^ The EQ-VAS score per interquartile range increment at moving average 0 to 3 days, except in the <30 years old age group (at lag1) adjusted for daily mean temperature, humidity, sex, age, education level, income, marital status, economic activity, smoking status, alcohol consumption, sleeping time, and the presence of chronic disease. PM_10_, particulate matter measuring <10 µm in diameter; NO_2_, nitrogen dioxide; CO, carbon monoxide; SO_2_, sulfur dioxide; CI, confidence interval.

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
