# Peer review of "Short-Term Impacts of Ambient Air Pollution on Health-Related Quality of Life: A Korea Health Panel Survey Study"

_ijerph, 2020, doi:10.3390/ijerph17239128_

Round 1

Reviewer 1 Report

This draft aimed to quantify short-term impacts of ambient air pollution on health-related quality of life in Korea, which is import and interesting. There are several comments as below.

Major comments:

  1. 5 and O3 are two most important pollutants, and I think the authors should add O3 analysis.
  2. There are many analysis about the health impacts of air pollution, however, the reviewers and readers can not judge whether the conclusions is right or not with the original data. So, is it possible to make the data and codes of research as supplement?

Specific comments:

  1. The authors should define ‘short-term’ in the beginning of the draft.
  2. More descriptions are necessary about ‘EQ-VAS’.

Author Response

This draft aimed to quantify short-term impacts of ambient air pollution on health-related quality of life in Korea, which is import and interesting. There are several comments as below. To activate the figure, please refer to the attached Word file.

  • The authors deeply appreciate a comprehensive review of the manuscript by the reviewer and for the valuable comments and suggestions. We have responded to the comments and the manuscript was revised accordingly.

Major comments

  1. PM5 and O3 are two most important pollutants, and I think the authors should add O3 analysis.
  • According to the reviewer’s comment, we analyzed the changes in EQ-VAS due to short-term exposure of ozone (O3). The daily average concentration of O3 was calculated as the highest 8-hour per day value by calculating the moving average level for 8 hours of 24 observational data measured every hour during the day.

Table 1. Exposure level of ozone and meteorological indices during the study period

Percentile

Mean

SD

Min

25th

50th

75th

Max

IQR

O3 (ppb)

27.9

11.3

2.1

20.0

28.6

35.2

79.7

15.2

SD, standard deviation; O3, ozone; IQR, interquartile range; ppb, parts per billion.

  • We analyzed with GAMM to observe the change of EQ-VAS for O3 exposure in subjects with chronic disease. However, it was not statistically significant.

<Figure 1. Change in the EQ-VAS scores per unit increment of ozone in subject with chronic diseases at lag0-3 days. The model was adjusted for sex, age, education, income, marital status, smoking, alcohol consumption, economic status, and sleeping time. CI, confidence interval.>

  • We analyzed all subjects stratified by sex and age as shown in Table 3 presented in the manuscript. The EQ-VAS did not decrease significantly with ozone exposure in contrast to the exposure to other air pollutants. Therefore, we did not present the results on the manuscript.

Table 2. The change of EQ-VAS scores per interquartile range increment of ozone in all subjects and subjects with chronic disease stratified by sex and age group.

Groups

EQ-VAS (95% CI)a

O3 (ppb)

All subjects

All

0.237 (-0.050-0.524)

With chronic disease

0.386 (0.012--0.760)

Sex

Male

All

-0.001 (-0.426-0.425)

With chronic disease

0.173 (-0.402-0.747)

Female

All

0.423 (0.035-0.810)

With chronic disease

0.559 (0.067-1.052)

Age group

(years)

<30

All

0.200 (-0.698-1.099)

With chronic disease

1.180 (-0.643-3.003)

30–39

All

0.139 (-0.759-1.038)

With chronic disease

0.084(-1.114-1.283)

40–49

All

0.327 (-0.245-0.899)

With chronic disease

0.274 (-0.566-1.114)

50–64

All

0.350 (-0.194-0.893)

With chronic disease

0.509 (-0.112-1.130)

≥65

All

0.065 (-0.585-0.715)

With chronic disease

0.281 (-0.399-0.961)

aThe EQ-VAS score per interquartile range increment at moving average 0 to 3 days, except the in <30 years aged group (at lag1) adjusted for daily mean temperature, humidity, sex, age, education level, income, marital status, economic activity, smoking status, alcohol consumption, sleeping time, and the presence of chronic disease. CI, confidence interval.

  1. There are many analyses about the health impacts of air pollution, however, the reviewers and readers cannot judge whether the conclusions is right or not with the original data. So, is it possible to make the data and codes of research as supplement?
  • Following the reviewer’s comments, we provided the dataset and code as a supplement.

Specific comments:

  1. The authors should define ‘short-term’ in the beginning of the draft.
  • In general, the exposure period of air pollution is classified into short-term and long-term exposure. Short-term exposure stands for acute exposure, a short contact with hazardous substances in the atmosphere, which lasts from hours to days. If such exposure lasts from months to years, it is defined as long-term exposure. We present this additional explanation in the introduction and added the reference related to short-term exposure to air pollution also been added (lines 54-60).
  1. More descriptions are necessary about ‘EQ-VAS’.
  • According to the reviewer’s comment, we added descriptions about EQ-VAS in the method and discussion section (lines 48-60, 236-251).

Reviewer 2 Report

The work is very good in terms of editorial and content.
The two things I think should be changed are:
Extended conclusions, which are now modest.
Explaining what EQ-VAS means? What does it mean for a person/population when the EQ-VAS is reduced by e.g: -026?

I am interested in why for people over 65 years old for PM10 and SO2, EQ-VAS increases?

Figure 1. Do I understand well that low concentrations of pollutants have a positive (positive) effect on EQ-VAS? Thus has a positive effect on human health?

Author Response

The work is very good in terms of editorial and content.

  • The authors deeply appreciate a comprehensive review of the manuscript by the reviewer and for the valuable comments and suggestions. We have responded to the comments and the manuscript was revised accordingly.

The two things I think should be changed are:

  1. Extended conclusions, which are now modest.
  • According to the reviewer’s comment, we revised the conclusion (lines 293-299).

  1. Explaining what EQ-VAS means? What does it mean for a person/population when the EQ-VAS is reduced by e.g: -026?
  • The visual analog scale (VAS), which is reliable for measuring the level of health because of its simplicity, ease of care, and suitability for repeated measurements [14], was used as a measure of HRQOL in the study of cancer patients by Priestman and Baum in the 1970s [15]. The HRQOL considers an individual’s perception of wellbeing, which should be regarded as an index of adverse health outcomes and an important indicator of morbidity and mortality during health interventions. (Lines 48-53)

Table 3 showed the results using GLMM after controlled all covariates according to the moving average exposure level of 0-3 days of NO2 in females with chronic diseases age 50 to 64 years old. We interpreted the reduction of EQ-VAS by -1.435 applying the IQR range of 13.8 ppb of NO2. However, one might be suggested that, based on these results, a reduction in EQ-VAS is necessary to compare the absolute risk levels such as morbidity or mortality.

Table 3. The results of GLMM in females with chronic disease in aged 50-64 years

Estimate

Std. Error

df

t value

Pr(>|t|)

(Intercept)

65.5981

2.8290

2809.1926

23.1870

< 2e-16

no2_ma_03

-0.1043

0.0288

3437.0198

-2.5840

0.0098

temp_ma_03

0.0430

0.0338

3562.4557

1.2740

0.2029

humid_ma_03

-0.0126

0.0194

3712.2543

-0.6510

0.5152

Age

-0.5568

0.1431

8590.0000

-3.891

0.000101

Education

1.9170

0.3482

1031.0815

5.5050

0.0000

Smoke

0.2020

0.9834

1684.9595

0.2050

0.8373

Income

0.7220

0.2251

2634.3507

3.2070

0.0014

Drinking

-0.4900

0.5917

3187.7772

-0.8280

0.4076

Marital status

0.1687

0.5242

1141.6014

0.3220

0.7476

Economic status

-0.0040

0.5742

2256.0862

-0.0070

0.9945

Chronic disease

-5.3549

0.7404

2630.6982

-7.2330

0.0000

Sleeping time

0.5683

0.1969

3567.8660

2.8860

0.0039

Here, the EQ-VAS can be explained in relation to the level of education (estimate value=1.917), which is similar to a decrease of 1.435. The decline in EQ-VAS due to NO2 exposure can be interpreted as being the same as the daily health conditions felt in a population with a lower education level. In addition, in can be interpreted that the economic level (estimate value=0.722) is similar to the level of daily health and life satisfaction felt by the populations of two lower level in economic status compared with the reduced EQ-VAS due to NO2 exposure. These compared results are not presented in manuscript, but can be interpreted with socio-economic factors, such as the questions mentioned in the reviewer’s comments.

  1. I am interested in why for people over 65 years old for PM10 and SO2, EQ-VAS increases?
  • In this study, the reason that the decrease in EQ-VAS according to exposure to air pollution was statistically significant in subjects <65 years old, because the exposure frequency of air pollution was higher due to higher outdoor activities than the subjects ≥65 years old (Shin, J.; Park, J.Y.; Choi, J. Long-term exposure to ambient air pollutants and mental health status: a nationwide population-based cross-sectional study. PLoS One. 2018, 13, e0195607.). However, in population with ≥65 years old, the EQ-VAS per IQR increment of PM10 and SO2 increased 0.116 (95% CI: -0.376-0.609) and 0.007 (95% CI: -0.626-0.639), respectively, but they were not statistically significant. This suggests that EQ-VAS in the population over 65 years of age is more affected from demographic and socioeconomic factors compared to the reduction due to exposure to air pollution. In fact, the result showed that the decrease in EQ-VAS was greatest in ≥65 years old as the economical level decreased. This means that the effects of EQ-VAS from exposure to air pollution are not significant in the elderly with relatively lower outdoor activity.
  1. Figure 1. Do I understand well that low concentrations of pollutants have a positive (positive) effect on EQ-VAS? Thus has a positive effect on human health?
  • In this study, we performed GAMM in subjects with chronic diseases excluding only the Jeju area. In PM10, NO2, and CO, excluding SO2, the EQ-VAS increased as the exposure level increased under a certain concentration, but it was confirmed that it was not statistically significant because the confidence intervals all overlap. Therefore, overall trend of decrease was assumed in the estimation of parameter in GLMM. However, in case of SO2, there was a increasing tendency of EQ-VAS at the level below 7 ppb. We interpreted it in the GLMM as an overall linear trend.

<Figure 2>: Please refer to a Word file attached.

Reviewer 3 Report

Interesting paper addressing an important issue. There are some points that I would like the authors to address:

  • Line 20: "..showed a significant change the EQ-VAS.." This line needs to be rephrased.
  • The authors must improve the quality of presentation, especially the equations. There is no numbering and also the alignment is not uniform.
  • Why did the authors use GLMM and GAMM model? Did they compare the performance with other models before choosing a specific model? It is always better to highlight the reason behind selecting a method for statistical analysis.
  • Line 128-130: What is beta?
  • Figure 1: Mention the units in the X-axis.
  • The Literature Review is very weak. I would suggest the authors to do a more extensive background study so that it is more clear what are the research gaps and how their work fills those gaps. It would be better to consider more recent works to get a better impression of the how ambient air quality impacts health related quality of life.
  • I am not convinced with the point that PM2.5 data is not considered in this study. When talking about ambient air pollution and its effect on human health, PM2.5 is much more dangerous as it can penetrate deep into the lungs. 
  • A thorough proofreading has to be done to improve the presentation of content as well as removing grammatical mistakes/typos. 

Author Response

Interesting paper addressing an important issue. There are some points that I would like the authors to address.

  • The authors deeply appreciate a comprehensive review of the manuscript by the reviewer and for the valuable comments and suggestions. We have responded to the comments and the manuscript was revised accordingly.

  1. Line 20: "..showed a significant change the EQ-VAS.." This line needs to be rephrased.
  • According to the reviewer’s comment, we corrected the sentence (line 20).
  1. The authors must improve the quality of presentation, especially the equations. There is no numbering and also the alignment is not uniform.
  • According to the reviewer’s comment, we entered the number in the formula (line in 130 and 134).
  1. Why did the authors use GLMM and GAMM model? Did they compare the performance with other models before choosing a specific model? It is always better to highlight the reason behind selecting a method for statistical analysis.
  • Generalized linear mixed model (GLMM) is a model extended from linear mixed model for the distribution of the dependent variable that does not fits the typical distribution. In the case of repeated measurements such as panel study, outcomes from the same object may correlate with each other, and these characteristics must be considered in the analysis. In addition, in this study, a GLMM was used to maximize the power and analyze the non-linear changes in EQ-VAS according to the exposure level of air pollution.

This model is an extended form of the generalized additive model, which is a regression model including non-linear functions, and is function that links several cubic functions localized by applying a natural cubic splines (NS) to air pollutants. It is widely used to describe a variable with a time-series change.

The Korea Health Panel data used in this study was repeated once a year by individuals. A mixed-effect model was used to control characteristics that do not change even with time changes in an individual, factors related to quality of life and exposure levels to air pollution that change over time.

  1. Line 128-130: What is beta?
  • Beta is general sign in the model formula used when presenting a research model, and it is the sign of the estimated values of each covariate after GLMM was performed in each model (line in 134).
  1. Figure 1: Mention the units in the X-axis.
  • According to the reviewer’s comment, we added the x-axis unit in Figure 1 in the revised version.
  1. The Literature Review is very weak. I would suggest the authors to do a more extensive background study so that it is more clear what are the research gaps and how their work fills those gaps. It would be better to consider more recent works to get a better impression of the how ambient air quality impacts health related quality of life.
  • The studies on morbidity and mortality related to air pollution have been actively conducted, but studies on HRQOL have been reported only recently, because it was not an easy-to-measure indicator. Also, previous studies have observed changes in HRQOL due to long-term exposure (several years..) to air pollution, and there have been few studies related to short-term exposure (lines 54-60).

However, the authors determined that it is appropriate to evaluate the short-term effects of air pollution rather the long-term effects of air pollution on EQ-VAS, which is measured once a day. The reason why there are few studies related to short-term effects is that first, there are few data that have been conducted for many years in a panel study on the questionnaire on EQ-VAS, which is a measure of HRQOL. The second is that few surveys provide information on survey dates. Therefore, there were limitations on adding more related literature. We have added more EQ-VAS as possible in discussion section (lines 279-292).

  1. I am not convinced with the point that PM2.5 data is not considered in this study. When talking about ambient air pollution and its effect on human health, PM2.5 is much more dangerous as it can penetrate deep into the lungs.
  • Reviewer’s comments are very important. However, the PM5 data provided by the Ministry of Environment, Korea, has been provided since 2013, so there was a limit in the use of data of the previous years. In fact, in previous studies, it was reported that the health damage of PM2.5 was greater than that of PM10, but in this study, the health-related quality of life was not evaluated due to limitations in the use of PM2.5 data. This is mentioned as a limitations of the discussion (lines 254-258).
  1. A thorough proofreading has to be done to improve the presentation of content as well as removing grammatical mistakes/typos.
  • According to the reviewer’s comment, we revised the manuscript.

Round 2

Reviewer 1 Report

The revised manuscript could be accepted.

Reviewer 3 Report

I would like to thank the authors for carefully answering all the comments and providing justification where it was needed. The paper can be presented in its present form.